Regulatory role of transcription factor c-Myc in the pathogenesis of psoriasis

Cao Yue
Niu Xu-Ping
Zhang Kai-Ming zhangkaiming@sina.com
Shanxi Key Laboratory of Stem Cells for Immunological Dermatosis, Institute of Dermatology, Taiyuan Central Hospital of Shanxi Medical University , Taiyuan , Shanxi , China
Jaremko Łukasz
Electronic publication date: 2025 Jul 22
Publication date: 2025
Volume: 13
Electronic Location ID: e19706
Received 2025 Jan 22; Accepted 2025 Jun 15
Copyright: ©2025 Cao et al.
Copyright year: 2025
Copyright holder: Cao et al.
License: This is an open access article distributed under the terms of the Creative Commons Attribution License, which permits unrestricted use, distribution, reproduction and adaptation in any medium and for any purpose provided that it is properly attributed. For attribution, the original author(s), title, publication source (PeerJ) and either DOI or URL of the article must be cited.
License URL: https://creativecommons.org/licenses/by/4.0/

Keywords: c-Myc, Psoriasis, Keratinocytes, Proliferation, Metabolic reprogramming

Funding: Taiyuan Bureau of Science and Technology, Science, Technology, and Innovation Program of National Regional Medical Center 202214 Fundamental Research Program of Shanxi Province 202203021222409 202203021211009 The work was supported by the Taiyuan Bureau of Science and Technology, Science, Technology, and Innovation Program of National Regional Medical Center (No. 202214) and Fundamental Research Program of Shanxi Province (No. 202203021222409, No. 202203021211009). The funders had no role in study design, data collection and analysis, decision to publish, or preparation of the manuscript.

==============================
Psoriasis is a chronic relapsing dermatosis characterized by hyperproliferation and poor differentiation of keratinocytes (KCs). The c-Myc gene is one of the main members of the Myc family and exerts multiple biological functions. C-Myc is highly expressed in psoriatic lesions. The co-expressed genes and coexisting factors of c-Myc determine the final survival of cells. The high expression levels of c-Myc in the skin lesions of psoriatic patients are associated with the continuous proliferation of KCs, and form an abnormal state of epidermal dynamics. C-Myc is also involved in the induction of metabolic reprogramming of cells in the development of psoriasis, thus exacerbating the excessive proliferation of psoriatic epidermis. In this review, we focus on the mechanisms of the transcription factor c-Myc in the pathogenesis of psoriasis and its clinical implications.

Introduction

Psoriasis is a chronic relapsing skin disease with worldwide prevalence of 2%–3%. Psoriasis seriously affects the physical and mental health and quality of life of patients (Barrea et al., 2016; Gisondi et al., 2018; Kim et al., 2019). One of the typical hallmarks of psoriasis is the excessive proliferation of keratinocytes (KCs), resulting from a feedback cycle of KCs through crosstalk with immune cells (Huang et al., 2019; Makuch et al., 2023; Zhang et al., 2018). The current treatment of psoriasis can not fundamentally solve the needs of patients, and this review can provide new ideas and strategies for the diagnosis and treatment of psoriasis. This review will be of true interest to the dermatologists and psoriasis patients because it covers the molecular and clinical scope of dermatology to better understand the possible pathogenesis and potential therapeutic targets of psoriasis.

The human c-Myc oncogene is located in the 4th band of the long arm 2 region of chromosome 8, containing three exons and two introns. Exon 1 plays a regulatory role, but does not participate in protein coding. The c-Myc protein is a protein with 439 amino acid residues encoded jointly by exon 2 and exon 3, with a molecular weight of 62 kDa, located in the nucleus. c-Myc contains three structural domains (Fig. 1) (Herbst et al., 2004): an N-terminal transcriptional activation domain (TAD), a non-specific DNA binding region, and a target sequence located at the C-terminal that binds to basic helix loop helix (bHLH) and leucine zipper (LZ). Among these domains, TAD is required to activate the expression of target genes, including two conserved domains, MBI and MBII (Herbst et al., 2004). The intermediate region regulates the turnover of c-Myc. bHLH plays an important role in the binding of c-Myc to specific DNA sequences (Blackwell et al., 1990), and the HLH-LZ domain of c-Myc can form heterodimers with Max, activating the transcription of related genes (Amati et al., 1993).

Figure 1 The structure of c-Myc.

C-Myc contains three structural domains: an N-terminal transcriptional activation domain (TAD), a non-specific DNA binding region, and a target sequence located at the C-terminal that binds to basic helix loop helix (bHLH) and leucine zipper (LZ) (Herbst et al., 2004).

The c-Myc gene is one of the main members of the Myc family and displays various biological functions. The c-Myc protein encoded by the c-Myc gene can promote cell proliferation, inhibit stem cell differentiation, promote angiogenesis, interfere with cell apoptosis, and induce cell metabolic reprogramming and other basic biological activities (Fig. 2) (Kenneth & White, 2009; Meyer & Penn, 2008; Ponzielli et al., 2005). Under normal physiological conditions, the expression levels of c-Myc are strictly regulated. When the cells are in quiescent phase, the cellular levels of c-Myc are relatively low. Growth factors upregulate expression of c-Myc, initiating the transcription of downstream target genes. Once the cell enters a quiescent phase, c-Myc expression returns normal. Ubiquitination of the transcription factor c-Myc is a necessary for its transcriptional activity, and c-Myc is degraded upon completion of transcription. Increased expression levels of c-myc are linked to various diseases such as tumors (Magid, Murphy & Galis, 2003). As a classical oncogene, c-Myc protein is highly expressed in many tumors and causes hyperproliferation in the development of tumors (Dang, 2012; Dang et al., 1999; Grifoni & Bellosta, 2015).

Figure 2 Biological function of c-Myc.

The c-Myc protein encoded by the c-Myc gene can promote cell proliferation, inhibit stem cell differentiation, promote angiogenesis, interfere with cell apoptosis, and induce cell metabolic reprogramming and other basic biological activities.

Biological functions of c-Myc

c-Myc regulates cell cycle

The ordered progression of the cell cycle in eukaryotes relies on a network regulatory system centered around cyclin-dependent kinases (CDKs), where cyclin is a positive regulator and cyclin-dependent kinase inhibitors (CKIs) are negative regulators. Together, these two factors regulate the kinase activity of CDKs and accurately coordinate the evolution of the cell cycle. C-Myc regulates the operation of the cell cycle at multiple levels. First, c-Myc acts as a transcription factor, directly regulating the expression of cell cycle regulatory factors such as CDKs, Cyclins and E2F transcription factors at the transcriptional level (Amati, Alevizopoulos & Vlach, 1998; Lutz, Leon & Eilers, 2002). Second, c-Myc activates the cyclin/CDK complex by CDK activating kinase (CAK) and cell division cycle 25 (Cdc25) phosphatase, ultimately achieving the transition of cells from G1 phase to S phase, and third, c-Myc not only blocks the transcription of cyclin dependent kinase inhibitor p21, but also induces Skp2 to regulate the ubiquitination degradation of p27 (Claassen & Hann, 2000). Thus, cells overexpressing c-Myc can break free from the limitations of cell cycle checkpoints, leading to cell deterioration and entering a state of uncontrolled proliferation (Bretones, Delgado & León, 2015).

Cell proliferation depends on the operation of cell cycle, while the continuous operation of cell cycle depends on the combination of cyclin dependent kinase and cyclin. Activation of c-Myc accelerates cell proliferation through regulation of G1/S transition of cell cycle. The increased expression of c-Myc can also indirectly activate cell cycle by attenuation of the effects of cell cycle inhibitors such as P21 and P27 (Chiyoda et al., 2012; Dwivedi et al., 2015; Tsai et al., 2014). Knockout of c-Myc can block cell proliferation and arrest cell cycle in cancer cells (Boddupally et al., 2012; Chen et al., 2014; Liu et al., 2017a). C-Myc protein can also increase the biosynthesis of mitochondria to maintain the requirement of high proliferation of transformed cells (Graves et al., 2012; Sarin et al., 2013).

c-Myc inhibits stem cell differentiation

The essence of cell differentiation results from selective gene expression. C-Myc regulates the differentiation of stem cells and maintains the versatile hematopoietic stem cells (Han et al., 2015; Li et al., 2011; Singh et al., 2015) and embryonic stem cells (Kidder, Yang & Palmer, 2008; Nie et al., 2012; Sabò et al., 2014). The ability of c-Myc to regulate the expression and function of stem cells is closely related to its carcinogenic activity. The expression levels of c-Myc are decreased during cell differentiation (Nie et al., 2012). During the differentiation process, a large number of genes must be turned on and off to achieve reprogramming of cell cycle. Pausing the amplification of c-Myc drivers will make reprogramming more effective and rapid. Afterwards, increased c-Myc levels strengthen the new cellular state. In embryonic stem cells, the high level of c-Myc enhances the undifferentiated state and prevents random differentiation (Nie et al., 2012).

c-Myc promotes angiogenesis

Neovascularization is an important nutritional guarantee for the survival of cells. Rapidly proliferating cells usually achieve growth and proliferation through neovascularization (Guerreiro et al., 2012; Quan et al., 2017; Schumann et al., 2014). Hypoxia inducible factor-1 (HIF-1α) promotes angiogenesis under hypoxia (Cui et al., 2012; Lyssiotis et al., 2012; Rankin & Giaccia, 2008), while c-Myc can induce HIF-1α expression, thus promoting angiogenesis. Conversely, downregulation of c-Myc and HIF-1α decreases expression of vascular endothelial growth factor and angiogenesis (Lee & Wu, 2015; Li et al., 2013; Li et al., 2016b).

c-Myc interferes with apoptosis

Apoptosis, a self-destruction mechanism, exists in cells and functions as immune surveillance. In normal cells, c-Myc can induce apoptosis. Askew et al. (1991) showed that the growth of myeloid leukemia cell lines depends on the cytokine IL-3. Both expression of the endogenous c-Myc gene and cell growth are inhibited in the absence of IL-3. Overexpression of c-Myc can remarkably induce cell apoptosis in IL-3 deficient cells. Because c-Myc can promote both cell proliferation and apoptosis, regulation of c-Myc in proliferation and apoptosis may be through different mechanisms. Because of tissue specificity, c-Myc regulates cell apoptosis via two mechanisms, i.e., Fas-FasL death receptor pathway and the mitochondrial pathway. In the Fas-FasL death receptor pathway, under the stimulation of apoptosis signals, Fas interacts with the death receptor Fas receptor, causing Fas associated protein with a novel death domain (FADD) to bind to the Fas receptor. Then, FADD recruits the precursor of caspase-8, which undergoes self-splicing activation and initiates apoptosis. c-Myc regulates FasL at the transcriptional level and prevents the downregulation of FasL, regulating the expression of Fas. In addition, c-Myc downregulates caspase-8 catalytic inactivation protein c-FLIP and acts on tumor necrosis factor (TNF), thereby promoting cell apoptosis involved in the death receptor pathway (Järvinen et al., 2011). In the mitochondrial pathway, c-Myc induces the release of cytochrome c from mitochondria into the cytoplasm. Subsequently, cytochrome c forms an apoptotic complex with apoptotic protease activating factor-1 (Apaf-1) and caspase-9. Activation of caspase-9 activates other caspases (such as caspase-3), ultimately inducing cell apoptosis (McMahon, 2014).

c-Myc induces cell metabolic reprogramming

Studies have also demonstrated that c-Myc regulates cell metabolism (Hartl & Bister, 2021; Hsieh et al., 2015). c-Myc not only regulates glutamine metabolism, but also directly binds to the promoters of key glycolytic genes, including glucose transporter 1 (GLUT1), hexokinase2 (HK2), phosphofructose kinase, phosphoglycerate kinase, α-Enolase and lactate dehydrogenase A (LDHA) to initiate their transcriptional expression, thereby promoting cell metabolic reprogramming. α-Enolase increases glycolysis and tumor cell proliferation (Dang, 2016; Hsieh et al., 2015), while LDHA is required to produce NAD+, a cofactor required for glycolysis to provide sufficient energy and material basis for tumor cell proliferation. Knockout of c-Myc in gastric cancer cells inhibits cell proliferation and glycolysis levels. Compared to knockout of either PKM2 or c-Myc alone, double knockout of PKM2 and c-Myc has a more significant inhibitory effect on gastric cancer cells (Dang, Le & Gao, 2009; Gao et al., 2019; Zhu et al., 2016). GLUT1, as another key molecule in glycolysis, is associated with tumor proliferation. In breast cancer, inhibition of c-Myc decreases translation and transcription of GLUT1 and inhibits growth of tumor cells (Chen et al., 2011; Hiscox & Nicholson, 2008; Jain et al., 2015). c-Myc also increases the variable splicing of pyruvate kinases by upregulating heterogeneous nuclear ribonucleoproteins (hnRNPs). c-Myc upregulates the transcription of pyrimidine binding proteins, hnRNPA1 and hnRNPA2, thereby maintaining a high proportion of PKM2/PKM1 and promoting glycolysis (Doherty et al., 2014; Gan et al., 2016; Wahlström & Henriksson, 2015).

C-Myc is the main oncoprotein that causes tumor cells to rely on glutamine. In low glucose and oxygen environments, c-Myc-induced glutamine metabolism is beneficial for cell survival. c-Myc-induced upregulation of glutamine synthase expression increases the conversion of glutamine to glutamate in the tricarboxylic acid cycle (Bott et al., 2015; Wang et al., 2019; Ye et al., 2018). In addition, c-Myc promotes the absorption of glutamine by increasing the expression of glutamine transporter sodium/glucose cotransporters and high affinity cationic amino acid transporters. c-Myc increases expression of glutamine transporter alanine serine cysteine transporter 2 (ASCT2). The latter transports glutamine into cells where glutamine is converted to glutamate by glutaminase (GLS). Moreover, nucleotide metabolism is regulated by the transcription factor c-Myc, which can bind to multiple important genes in the nucleotide metabolism pathway and directly regulate the metabolic enzymes, such as carbamoyl phosphate synthase. In addition, serine hydroxymethyltransferase 1 (SHMT1) and SHMT2 provide one carbon unit to tetrahydrofolate, which is methylated to deoxyuridine by thymidylate synthase. SHMT2 is an enzyme involved in one carbon metabolism and is crucial for the synthesis of deoxyribonucleoside triphosphate. c-Myc can directly activate SHMT1 and SHMT2, leading to abnormal nucleotide metabolism. A recent study showed that overexpression of c-Myc enhances eukaryotic translation initiation factor 4E (eIF4E), driving the translation of phosphate ribose pyrophosphate synthase 2 (PRPS2) to promote nucleotide synthesis (Cunningham et al., 2014; Mannava et al., 2008; Miao & Wang, 2019).

Additionally, c-Myc also affects the lipid metabolism reprogramming of tumor cells by regulating genes related to fatty acid metabolism, including ATP citrate lyase, acetyl CoA carboxylase α, stearoyl CoA desaturase and fatty acid synthase (FASN) (Jia et al., 2020; Li et al., 2016a; Liu et al., 2017b). Prostate epithelial cells express high levels of c-Myc, which enhances the transcription and translation levels of FASN. Metabolomics analysis also showed the enrichment of phospholipids and citrate lipid precursors in c-Myc overexpressing cells, suggesting that high levels of c-Myc are associated with increased fatty acid synthesis. In summary, c-Myc plays a crucial role in the biological processes of tumor cells by promoting various pathways, including glycolysis, glutamine decomposition, nucleotide synthesis and fatty acid metabolism.

The role of c-Myc in psoriasis

The excessive proliferation of keratinocytes in psoriasis is similar to that of tumor cells. Evidence suggests the involvement of c-Myc in the development of psoriasis (Ba et al., 2016; Moses et al., 2016; Yang et al., 2012). c-Myc has dual effects on cells, i.e., stimulation of cell proliferation and promotion of cell apoptosis (Chiarugi & Ruggiero, 1996; Greider et al., 2002; Sharma, Patel & Srikant, 1996). In normal skin, c-Myc stimulates the proliferation of KCs, but its effect on apoptosis is inhibited by co-expression of bcl-2. However, the expression levels of bcl-2 protein are decreased in basal cells of psoriatic lesions (Ba et al., 2016). Thus, down-regulation of bcl-2 initiates c-Myc-induced apoptosis, resulting in the apoptosis of hyperproliferated KCs in psoriasis. At the same time, the overexpression of c-Myc attenuates the growth arrest effect of p53 on the cell cycle, resulting in the cell bypass the G1/S checkpoint of the cell cycle, consequently inducing mitotic imbalance, while the abnormal mitotic signal can further induce cell apoptosis (Chiarugi & Ruggiero, 1996). This line of evidence indicates a crucial role of c-Myc protein in proliferation and apoptosis of psoriatic KCs. Because of the increased mad1 mRNA and c-Myc expression in psoriasis and wound epidermis (Miyoshi et al., 2011; Nakajima et al., 2019; Sano et al., 2005; Bedini et al., 2007; Li et al., 2007; Zhang et al., 2008), targeting either mad1 mRNA and c-Myc can be beneficial for psoriasis.

The c-Myc gene coding products are mainly involved in the regulation of cell division, differentiation and growth cycle (Eischen et al., 2001; Patel & McMahon, 2006; Patel & McMahon, 2007). Under normal conditions, cells do not express or only express low levels of c-Myc, while low levels of c-Myc inhibit cell proliferation, induce cell differentiation, and main KC proliferation and differentiation at the normal levels. High expression levels of c-Myc can drive the cells away from normal growth, causing hyperproliferation (Gopal et al., 1992; Hulette et al., 1992; Pedrazzoli et al., 1989). The high expression levels of c-Myc in the skin lesions of psoriatic patients contribute to hyperproliferation of KC, and formation of an abnormal state of epidermal dynamics.

The hyperkeratosis and parakeratosis observed in psoriasis are direct manifestations of disrupted epidermal differentiation and barrier function. The core mechanism involves abnormal proliferation and differentiation of keratinocytes, driven by the inflammatory microenvironment (Albanesi et al., 2018). Under physiological conditions, keratinocytes undergo a balanced proliferation-apoptosis cycle, completing their differentiation from the basal layer to the stratum corneum over approximately 28 days. In psoriasis, dysregulated c-Myc overexpression alters the keratinocyte cycle, shortening it to 3-5 days. This results in the rapid accumulation of immature keratinocytes in the upper epidermal layers, ultimately leading to pathological thickening of the stratum corneum.

In psoriasis, immune cells such as Th17 cells interact with keratinocytes, forming a self-reinforcing pro-inflammatory and hyperproliferative feedback loop. Immune cells release inflammatory cytokines (e.g., IL-17, IL-22, TNF-α), which activate proliferative signaling pathways (e.g., STAT3 and MAPK) in keratinocytes (Griffiths et al., 2021). In response, keratinocytes secrete chemokines such as CXCL8 and CCL20, recruiting additional immune cells to infiltrate the skin and amplifying inflammation. Studies have shown that the binding of interleukin-22 (IL-22) to the KC-specific receptor IL-22R1 induces the activation and phosphorylation of signal transducer and activator of transcription 3 (STAT3), the key factor of signal pathway, and causes the overexpression of the cell cycle regulator (c-Myc), thus altering KC cycle and provoking abnormal proliferation and keratinization of the non-proliferative upper layer and granular layer of the epidermis, leading to the development of psoriatic lesions (Cai et al., 2016; Cai et al., 2017; Mullan et al., 2013).

In tumor cells, c-Myc can induce glycolysis, thus promoting the metabolic reprogramming of cells (Brooks & Hurley, 2009; Prochownik, 2004; Sun & Hurley, 2009). Cell proliferation rate is linked to metabonomic characteristics and activity (Boehncke & Schön, 2015; Griffiths & Barker, 2007). At present, aerobic glycolysis has been widely recognized as an important symbol of tumors. Almost all tumors have been confirmed to have the “Warburg effect”, that is, tumor cells still use glycolysis as the main energy supply mode even under the condition of sufficient oxygen (Hanahan & Weinberg, 2011; Meehan & Vella, 2016; Orgaz, Herraiz & Sanz-Moreno, 2014). Glycolysis plays a key role in the large-scale biosynthesis process required for the proliferation of active tissues (Ganapathy-Kanniappan, 2018; Vander Heiden, Cantley & Thompson, 2009; Yu et al., 2015). Compared with mitochondrial aerobic respiration, glycolysis is a fast way to produce energy, and causes accumulation of a large number of intermediate metabolites during cell glycolysis. Metabolites can be used as raw materials for macromolecules and organelles to assemble new cells through a variety of biosynthetic pathways (Vander Heiden, Cantley & Thompson, 2009).

Our previous studies showed that the energy metabolism of KCs in psoriatic lesions is similar to that of tumor cells. Psoriatic KCs display a higher potency in metabolic reprogramming and stimulation of dermal mesenchymal stem cell proliferation, possibly contributing to the pathogenesis of psoriasis (Cao et al., 2022); Psoriasis derived dermal mesenchymal stem cells can increase glycolysis levels in normal keratinocytes (Li et al., 2020). Compared with normal dermal mesenchymal stem cells, the levels of glycolysis are significantly increased in psoriatic dermal mesenchymal stem cells (Wang et al., 2020). This bulk of evidence indicates that energy metabolism is altered in the lesional skin of psoriasis and that excessive dependence on glycolysis metabolism results in a rapid production of a large amount of adenosine triphosphate (ATP), which provides energy and raw materials to support excessive cell proliferation. Therefore, we speculate that c-Myc involves in the induction of cell metabolism reprogramming in the development of psoriasis, thus aggravating the excessive proliferation of psoriatic epidermis (Fig. 3) (Cao et al., 2022). Hower, further studies are need to validate this speculation.

Figure 3 The role of c-Myc in psoriasis.

The binding of IL-22 to the KC-specific receptor IL-22R induces the activation and phosphorylation of STAT3, and causes the overexpression of c-Myc, leading to excessive synthesis of GLUT1, resulting in an increased glycolysis and the induction of cell metabolism reprogramming in the development of psoriasis. Consequently, altered metabolism can cause the hyperproliferation of KC and abnormal proliferation of psoriatic epidermis.

Current treatment approaches for psoriasis are categorized into topical therapy and systemic therapy (Mason et al., 2013). Mild to moderate psoriasis is managed with topical treatments, where various therapeutic agents such as calcineurin inhibitors, vitamin D analogs, and corticosteroids are applied to affected skin via conventional formulations like gels and ointments. However, these delivery vehicles (gels, ointments, etc.) fail to effectively penetrate the outermost skin barrier, hindering drug delivery to deeper dermal layers and reducing therapeutic efficacy (Paudel et al., 2010). Additionally, these carriers have the disadvantages of high viscosity, slow skin penetration, and poor patient compliance, which limit the application of local treatment. Systemic therapy is primarily used for moderate to severe psoriasis. Traditional agents like acitretin, cyclosporine, and methotrexate exhibit limitations such as slow onset, suboptimal efficacy, poor tolerability, and potential severe toxicities affecting organs like the liver and kidneys (Roenigk Jr, Fowler-Bergfeld & Curtis, 1969). In recent years, the widely used biological agents including TNF-α inhibitor, IL-17 inhibitor, IL-23 inhibitor and IL-12/23 inhibitor can produce good therapeutic effect, but they need to be administered subcutaneously and intravenously to achieve the effect of systemic treatment. Nevertheless, this delivery approach causes patient discomfort, necessitates professional healthcare involvement, and may impose financial burdens due to high costs. Consequently, the development of convenient and effective therapeutic strategies for psoriasis remains an urgent unmet need. We propose that future innovations targeting c-Myc inhibitors, gene therapies, or metabolic interventions may emerge as promising approaches for transformative psoriasis management.

Survey Methodology

We search through search engines (such as Google Academic, CNKI), literature databases (such as PubMed, Web of Science) and academic journals. In addition, we also pay attention to the selection of keywords (including c-Myc; psoriasis; keratinocytes; proliferation; metabolic reprogramming) and evaluate the quality and authority of the literature to ensure the comprehensiveness and impartiality of the review.

Conclusions

C-Myc exhibits biological functions. More and more evidence indicates the crucial role of c-Myc in regulating epidermal proliferation in psoriasis. However, the underlying mechanisms whereby c-Myc is involved in the pathogenesis of psoriasis and other hyperproliferative disorders are still unclear. Moreover, whether overexpression of c-Myc is the cause or the result of hyperproliferative disorders remains to be elucidated. Although evidence suggests that targeting c-Myc signaling pathway can be a novel approach for the management of psoriasis and possibly other hyperproliferative disorders, further studies are needed to determine the clinical significance of c-Myc signaling pathway in clinical setting.

Abbreviations

KCs keratinocytes

TAD transcriptional activation domain

bHLH basic helix loop helix

LZ leucine zipper

CDKs cyclin-dependent kinases

CKIs cyclin-dependent kinase inhibitors

Cdc25 cell division cycle 25

HIF-1α Hypoxia inducible factor-1

FADD Fas associated protein with a novel death domain

TNF tumor necrosis factor

Apaf-1 apoptotic protease activating factor-1

GLUT1 glucose transporter 1

HK2 hexokinase2

LDHA lactate dehydrogenase A

hnRNPs heterogeneous nuclear ribonucleoproteins

ASCT2 alanine serine cysteine transporter 2

SHMT1 serine hydroxymethyltransferase 1

eIF4E eukaryotic translation initiation factor 4E

PRPS2 phosphate ribose pyrophosphate synthase 2

FASN fatty acid synthase

IL-22 interleukin-22

STAT3 signal transducer and activator of transcription 3

Additional Information and Declarations

Competing Interests

Author Contributions

Data Availability

The authors declare there are no competing interests.

Yue Cao conceived and designed the experiments, performed the experiments, prepared figures and/or tables, and approved the final draft.

Xu-Ping Niu performed the experiments, prepared figures and/or tables, authored or reviewed drafts of the article, and approved the final draft.

Kai-Ming Zhang analyzed the data, authored or reviewed drafts of the article, and approved the final draft.

The following information was supplied regarding data availability:

This is a literature review.

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
