# Peer review of "Regulatory role of transcription factor c-Myc in the pathogenesis of psoriasis"

_PeerJ, doi:10.7717/peerj.19706_

## Round 0.1 · original submission · Major Revisions

The manuscript provides a thorough review of c-Myc’s role in psoriasis pathogenesis but would benefit from a deeper discussion on the therapeutic implications of targeting c-Myc, the direct (if possible at all) and indirect targeting (please mention several potential targets and novel achievements). Expanding on current psoriasis treatments, their limitations, and potential c-Myc-targeted strategies (such as inhibitors or gene therapy) could increase the novelty and translational value of the review. Additionally, the authors should clarify the coexisting factors that interact with c-Myc in cell proliferation and apoptosis, and address the role of c-Myc in keratinization and inflammatory responses in keratinocytes. This would strengthen the manuscript and make it more clinically relevant.

Reviewer 1 ·

Basic reporting

The manuscript provides a clear and well-structured overview of the biological functions of c-Myc and its role in psoriasis pathogenesis. The language is professional and unambiguous, and the review is supported by relevant literature. However, the novelty of the article is somewhat lacking. While the authors comprehensively describe the biological functions of c-Myc and its involvement in psoriasis, the discussion does not sufficiently extend to the therapeutic implications of targeting c-Myc in psoriasis treatment.

To enhance the impact of the review, I suggest the authors expand their discussion on current therapeutic limitations for psoriasis and emphasize the potential of c-Myc as a therapeutic target. Specifically, a comparative analysis of existing psoriasis treatments and their shortcomings (such as biologics, small-molecule inhibitors, and systemic therapies) could help establish the clinical relevance of c-Myc inhibition. Furthermore, discussing potential c-Myc-targeted strategies, including inhibitors, gene therapy, or metabolic interventions, would improve the manuscript’s novelty and provide a more translational perspective.

Addressing these points would make the review more valuable to both researchers and clinicians, reinforcing the potential role of c-Myc in future therapeutic approaches for psoriasis.

Experimental design

As this is a review article, the traditional study design criteria do not apply.

Validity of the findings

The references cited in the manuscript are generally relevant and provide strong support for the biological functions of c-Myc and its role in psoriasis. However, most of the references focus on molecular and cellular mechanisms rather than clinical dermatology. Given that psoriasis is a chronic immune-mediated skin disease with a significant clinical burden, it is crucial to integrate more discussion on clinical aspects to enhance the translational value of the review.

Additional comments

None

Reviewer 2 ·

Basic reporting

The manuscript provides a comprehensive review of the role of the transcription factor c-Myc in the pathogenesis of psoriasis. The authors offer an in-depth analysis of c-Myc’s function in cell proliferation, apoptosis, metabolic reprogramming, and related processes. However, several concerns should be addressed:
The authors state, "The genes co-expressed with c-Myc and the coexisting factors determine the fate of cells. If the 213 coexisting factors stimulate c-Myc to promote cell proliferation, it is manifested as cell 214 proliferation" (lines 212–215). This statement is somewhat superficial. The authors should provide more detailed information regarding the coexisting factors. Specifically, they should identify which factors interact with c-Myc to drive cell proliferation and, similarly, which factors contribute to apoptosis.

Psoriasis is a skin disease also characterized by aberrant keratinization. It would be valuable to discuss the role of c-Myc in the keratinization process of keratinocytes (KCs) and how its dysregulation may contribute to disease pathology.

Additionally, exploring the role of c-Myc in the inflammatory response within KCs would further enhance the manuscript. Including this aspect would provide a more comprehensive understanding of c-Myc’s involvement in psoriasis.

Experimental design

no comment

Validity of the findings

no comment

---

## Round 0.2 · accepted · Accept

As the reviewers are satisfied with the current version and I also find the manuscript ready for publication, I would like to congratulate you on a job well done.

Thank you for your efforts, and I wish your work great visibility and impact.

Reviewer 1 ·

Basic reporting

no comment

Experimental design

no comment

Validity of the findings

no comment

Additional comments

Authors had made sufficient revisions to address my comments, and I recommend accepting the manuscript.

Reviewer 2 ·

Basic reporting

The authors have addressed my concerns to my satisfaction.

Experimental design

The review is comprehensive.

Validity of the findings

The article is comprehensive.